# Development of a Nucleic Acid Lateral Flow Immunoassay for the Detection of Human Polyomavirus BK

**DOI:** 10.3390/diagnostics10060403

**Published:** 2020-06-12

**Authors:** Yi-Huei Huang, Kuan-Yi Yu, Shou-Ping Huang, Hui-Wen Chuang, Wen-Zhi Lin, Juin-Hong Cherng, Yao-Wen Hung, Ming-Kung Yeh, Po-Da Hong, Cheng-Che Liu

**Affiliations:** 1Biomedical Engineering Program, Graduate Institute of Applied Science and Technology, National Taiwan University of Science and Technology, Taipei 10607, Taiwan; d10222201@mail.ntust.edu.tw; 2Graduate Institute of Life Sciences, National Defense Medical Center, Taipei 11490, Taiwan; 806302029@mail.ndmctsgh.edu.tw (K.-Y.Y.); 606020004@mail.ndmctsgh.edu.tw (H.-W.C.); mkyeh2004@gmail.com (M.-K.Y.); 3Department of Physiology and Biophysics, Graduate Institute of Physiology, National Defense Medical Center, Taipei 11490, Taiwan; 607020007@mail.ndmctsgh.edu.tw; 4Institute of Preventive Medicine, National Defense Medical Center, Taipei 11490, Taiwan; takashi@mail.ndmctsgh.edu.tw (W.-Z.L.); yaowen3@hotmail.com (Y.-W.H.); 5Department and Graduate Institute of Biology and Anatomy, National Defense Medical Center, Taipei 11490, Taiwan; i72bb@mail.ndmctsgh.edu.tw; 6Department of Gerontological Health Care, National Taipei University of Nursing and Health Sciences, Taipei 11219, Taiwan

**Keywords:** BK polyomavirus, lateral flow immunoassay, nucleic acid testing, point-of-care testing

## Abstract

The BK virus (BKV) is an emerging pathogen in immunocompromised individuals and widespread in the human population. Polymerase chain reaction is a simple and highly sensitive method for detecting BKV, but it is time consuming and requires expensive instruments and expert judgment. The lateral flow assay, a rapid, low-cost, minimal-labor, and easy-to-use diagnostic method, was successfully applied for pathogen detection. In this study, we used oligonucleotide probes to develop a simple and rapid sandwich-type lateral flow immunoassay for detecting BKV DNA within 45 minutes. The detection limit for the synthetic single-stranded DNA was 5 nM. The specificity study showed no cross-reactivity with other polyomaviruses, such as JC virus and simian virus 40. For the *Escherichia coli* containing BKV plasmid cultured samples, the sensitivity was determined to be 10^7^ copies/mL. The approach offers great potential for BKV detection of various target analytes in point-of-care settings.

## 1. Introduction

BK polyomavirus, also known as the BK virus (BKV), is a small (40–45 nm in diameter), icosahedral and nonenveloped virus that contains an approximately 5-kbp circular double-stranded DNA. This virus is a member of the Polyomaviridae family and the *Betapolyomavirus* genus [1,2]. Serological studies have reported that more than 90% of the global human population has tested positive for BKV [3,4,5,6]. Following primary infection, BKV remains as a latent infection in the epithelium of the renourinary tract [7,8]. BKV reactivates and replicates when individuals are immunosuppressed or immunocompromised due to cancer, pregnancy, diabetes, HIV infection, or transplants. Following renal or hematopoietic stem cell transplantation under an immunosuppressed condition, the reactivation of BKV may cause serious complications, including BKV-associated nephropathy (*BKVAN*), ureteric stenosis, and hemorrhagic cystitis [9,10,11,12]. The introduction of new and more potent immunosuppressants to clinical practices likely explains the high incidence of BKV-associated diseases [13,14].

In early BKV reactivation, viral shedding can be detected in the urine; furthermore, decoy cells, which form when renal tubular epithelial cells infected with BKV are shed in the urine, can be found in urinary sediments. Decoy cells have intranuclear viral inclusion bodies, and nuclear enlargement typifies their appearance. As BKV multiplies, the infection spreads from the renal medulla to the cortex, and BKV crosses into the peritubular capillaries, causing viremia and eventually leading to various tubulointerstitial lesions and BKVAN. The outcome depends on the degree of tissue damage, inflammation, and fibrosis. In severe cases, graft loss occurs [15,16]. There is no antiviral drug with strong clinical evidence supporting its effectiveness against BKV. Reconstructing patients’ BKV-specific immunity through immunosuppression reduction has been considered an effective treatment strategy to effectively clear the virus and maintain stable allograft function. Therefore, the monitoring of BKV replication and the appropriate adjustment of immunosuppression regimens are crucial to improving the control of BKV-associated diseases [17,18,19]. The use of polymerase chain reaction (PCR) to quantify viral DNA in the urine and plasma is standard in the clinical monitoring of BKV reactivation. Urine cytology can also be employed to detect decoy cells. However, the presence of BKV in the urine or blood is not related to renal impairment. Therefore, the gold standard for the diagnosis of BKVAN is renal transplant biopsy [16,20,21].

Although PCR is a simple and highly sensitive detection method, it is time consuming and requires expensive instruments and expert judgment. Hence, the development of easier-to-use and more economical complementary diagnostics is necessary for the improved monitoring of BKV reactivation in patients, particularly in laboratory settings or in areas with limited resources. Lateral flow assay (LFA) is an affordable detection method that is quick, simple, portable, user friendly, and easy to interpret. LFAs have been applied in many domains, including urinalysis, pathogen detection, veterinary medicine, quality control, food safety applications, environmental monitoring, biothreat detection, and drug rehabilitation [22]. Depending on the elements of recognition used, LFAs can be categorized as an antibody-based lateral flow immunoassay (LFIA), which uses a combination of antigens and antibodies, and nucleic acid lateral flow immunoassay (NALFIA), in which a single-stranded target analyte (DNA or RNA) forms a complex through complementary probe hybridization, generating detectable visual signals [23]. This study aimed to develop a simple and rapid sandwich-type LFIA for detecting BKV DNA. To verify the effectiveness of this method, gold nanoparticles (AuNPs) were used as an indicator, and the sequence of a short fragment of BKV was used as the model analyte. The analysis was completed within 45 min without nucleic acid amplification, thus verifying the feasibility of using the proposed method for detecting BKV when resources are scarce.

## 2. Materials and Methods

### 2.1. Target Area Detection and Probe Design

Data were collected from the National Center for Biotechnology Information (NCBI) database. As of November 2016, we had obtained completely coded DNA sequences of all genotypes: BKV (232 sequences), JC polyomavirus ((JCV) 464 sequences), and simian virus 40 ((SV40) 34 sequences). Lasergene 8.0 (DNASTAR, Madison, WI, USA) was used for multiple sequence alignment. First, we identified the consensus region in which the different sequences of each virus matched completely. Second, we conducted nucleotide sequencing of a segment containing more than 40 bases in the consensus region of each virus at the same sequence position in the genome. Subsequently, we numbered the variable positions at which the type of nucleotide at the same sequence position in JCV and SV40 simultaneously differs from that in BKV. The region with the highest number of variable positions was selected as the target area for detection. Vector NTI Advance 11.0 (Thermo Fisher Scientific, Waltham, MA, USA) was employed for the design of our probes. Two public databases, the NCBI and the DNA Data Bank of Japan, were used to compare and verify the specificity and homology of the screened target sequence fragment and the probe sequence. All the oligonucleotide probes and targets were purified through polyacrylamide gel electrophoresis and obtained from Genomics BioSci & Tech Co., Ltd. (New Taipei City, Taiwan). The detector probes (DP) and capture probes (CP) were thiolated and biotinylated, respectively. Designation of the sequences for each oligonucleotide are listed in Table 1.

### 2.2. Preparation of AuNP–DP Conjugates

The preparation of DP-functionalized AuNPs was adapted from a similar method in the literature [24]. All procedures were conducted in the dark. Two types of DP were used in this study: They were the reactive thiol anchor incorporated at the 3’ or 5’ terminus of the probe sequence, represented as 3’S and 5’S, respectively. We added 10 μL of the 100 μM thiol-modified oligonucleotide to 500 μL of 40 nm AuNP solutions (optical density (OD) at 525 nm = 3.2; Taiwan Advanced Nanotech, Taoyuan City, Taiwan), which were then incubated at room temperature for 16 h. The concentration of Tween 20 was brought to 0.4% and incubated for 1 h; thereafter, 0.1 M phosphate buffer (PB: 0.2 M NaH_2_PO_4_/Na_2_HPO_4_, pH 7.4) was added to the mixture to reach a final concentration of 10 mM. The solutions were gradually added to 50 mM NaCl through the dropwise addition of the 2 M NaCl solution. After standing for 8 h, this process was repeated with one more increment of 50 mM to reach a final concentration of 0.1 M. The excess of the reagents was removed through centrifugation at 8000 rpm for 30 min at 4 °C. The supernatant was discarded, and the red oily pellets were resuspended in 0.3 M NaCl with 10 mM PB (pH 7.4) before being centrifuged; the supernatant was then removed. The AuNP–DP was blocked in a 5% bovine serum albumin (BSA) solution for 1 h and subsequently centrifuged prior to having the supernatant removed. The AuNP–DP was then resuspended in 0.1 M phosphate-buffered saline (PBS, pH 7.4). The OD value of each resulting AuNP–DP solution was measured through ultraviolet (UV)–VIS spectrophotometry at 525 nm (E-Chrom Tech, Taipei, Taiwan). The color change of the solutions was recorded using a Nikon COOLPIX 4500 digital camera (Nikon, Tokyo, Japan) and characterized using UV–VIS spectrophotometry (E-Chrom Tech) to obtain the extinction spectra at 400–800 nm.

### 2.3. Fabrication of Lateral Flow Strip

The lateral flow strip consisted of two components: (1) An HF120 nitrocellulose membrane (EMD Millipore, Burlington, MA, USA) with the test and control lines and (2) a CF4 absorbent pad (GE Healthcare Life Sciences, Marlborough, MA, USA). For the preparation of the test zone, rabbit polyclonal anti-streptavidin antibody (4 mg/mL; GenScript, Piscataway, NJ, USA) and rabbit polyclonal anti-BSA antibody (5 mg/mL; Nordic-MUbio, Susteren, The Netherlands) were dispensed at a rate of 2 μL/cm on to the nitrocellulose membrane (2.5 cm × 30 cm) to generate the test and control lines, respectively; this was done using a BioJet Quanti 3000 dispenser (BioDot, Irvine, CA, USA) installed on an XYZ3060 platform (BioDot). The two lines were positioned at 0.5 cm apart. After being dried for 30 min at room temperature, the membrane was cut into strips a 0.5-cm-wide by CM4000 Guillotine Cutter (BioDot). Subsequently, strips were blocked in 1% (*w*/*v*) low-melting-point polyvinyl alcohol (in 20 mM Tris/HCl, pH 7.4) for 30 min at room temperature, washed once with sterilized deionized–distilled water, dried, immersed in 5% sucrose solution, and then dried again. Strips were stored in vacuum-sealed aluminum bags at 4 °C for further detection. Before use, the absorbent pad (4.0 cm × 0.5 cm) was pasted on the membrane with an overlap of approximately 0.2 cm.

### 2.4. Bacterial Culture and Nucleic Acid Extraction

Bacterial stock containing the pBKV (34-2) plasmid (ATCC 45025) was purchased from the American Type Culture Collection (Manassas, VA, USA) and cultivated overnight at 37 °C on lysogeny broth (LB) agar containing 50 μg/mL ampicillin. A single colony was transferred and incubated in LB medium containing 50 μg/mL ampicillin at 37 °C with shaking at 250 rpm. The reading of OD at 600 nm was determined through UV–VIS spectrophotometry (E-Chrom Tech) to estimate the cell density of bacteria. An OD at 600 nm corresponds to 2 × 10^8^ cells/mL [25]. Plasmid DNA was extracted from this bacterial culture using a QIAGEN Plasmid Maxi Kit (Qiagen, Venlo, The Netherlands) according to the manufacturer’s instructions and dissolved in sterilized deionized–distilled water. The sample was stored at −20 °C for future use.

### 2.5. Optimization, Sensitivity, and Specificity Assays

The optimization work proceeded in two phases. The first phase was the adjustment of the optimization quantity of the AuNP–DP and CP. Different amounts of AuNP–DP (1 OD, 2 OD, and 3 OD) and different concentrations of CP (50 μM, 100 μM, and 200 μM) were prepared; this was done to evaluate the effects of the strip on relative signal intensity in the presence of varying concentrations of target DNA (50 nM, 100 nM, and 1000 nM). The second part involved optimization assays of the modified position; this was done to evaluate the effects of the terminus positions of thiol and biotin incorporated at the probe sequence on relative signal intensity. In addition to 3’S and 5’S AuNP–DP, CP-incorporated biotin at the 3’ or 5’ terminus of the probe sequence was also used, which was represented as 3’B and 5’B, respectively. AuNP–DP and CP were paired to form four combinations: 3’S-5’B, 3’S-3’B, 5’S-5’B, and 5’S-3’B. Strips were then used for tests and analyses. Probe specificity was evaluated by synthesizing the sequence of the same target area segment in BKV, JCV, and SV40. Sensitivity was analyzed using eight-point, serially diluted, target DNA samples, with concentrations ranging from 5 nM to 1500 nM for evaluation. The practicability of the developed lateral flow strip was assessed and verified using 1 × 10^7^–10^10^ copies/mL samples of plasmid DNA containing the complete BKV (Dunlop) genome sequence. The data were fitted to a linear regression calibration curve using Origin 6.0 scientific data analysis and graphing software (OriginLab, Northampton, MA, USA); the curve represents for mean relative intensity from each dilution versus concentration of DNA of each dilution.

### 2.6. Assay Procedure

All hybridizations were performed in a thermal plate shaker incubator (CLUBIO, Taoyuan City, Taiwan) with a heated lid. We preheated 20 μL of the DNA sample at 95 °C for 5 min, added 1 μL of CP, and incubated it at 49 °C for 10 min. Then, we added 50 μL of AuNP–DP before incubating it for another 10 min. After the mixture was cooled to 40 °C, the solution was mixed with 1 μL of 1 mg/mL streptavidin (in Gibco 1X PBS, pH 7.4, Thermo Fisher Scientific) and incubated for 5 min. The strip was dipped into the solution, and the color of the test and control lines could be visualized within 10 min. The strip was washed once with 0.1 M PBS (pH 7.4) for 5 min and dried. The digital images of the resultant strip were recorded with a Nikon COOLPIX 4500 digital camera (Nikon). We then analyzed the relative intensity using Quantity One 1-D analysis software v4.4.0 (Bio-Rad Laboratories, Hercules, CA, USA). All images were converted into grayscale, and we created a volume box around the band, which was to be quantified using the Volume Rectangle Tool command in the software. The mean intensity of the pixels inside the volume boundary was obtained. For all intensity measurements, the background signal was measured in an adjacent area and subtracted. The relative intensity presented in the graph represents the ratio of the mean intensity of the band of the test line divided by the band of the control line.

### 2.7. Statistical Analysis

Data are presented as the mean ± standard deviation (SD). Statistical analyses were performed in GraphPad Prism (version 6.0; GraphPad Software, San Diego, CA, USA) using one-way and two-way analysis of variance accompanied by Fisher’s least-significant difference post hoc test. A *p* value < 0.05 indicated statistical significance. All experiments were repeated at least thrice, with the number of repetitions indicated in the text.

## 3. Results

### 3.1. Principle of NALFIA Measurement and Probe Design

Figure 1 illustrates the working principle of DNA measurement on the developed strip. When a sample solution contains target DNA, a pair of DNA probes in the solution is complementary to different positions in the sequence of the target DNA, thus forming hybridization complexes (AuNP-DP-target DNA-CP), and biotinylated CP subsequently binds to streptavidin. When the strip is soaked in the sample solution, the solution migrates along the strip and reaches the test line, after which the hybridization complexes are captured by anti-streptavidin antibodies on the membrane because of the streptavidin on the complex. The complexes then sediment, and a red band appears as they accumulate. The intensity of the test line is proportional to the quantity of target DNA in the sample. When the sample solution continues to migrate to the control line, excess AuNP–DP is captured by anti-BSA antibodies and sediments, resulting in a second red band. The assay is considered valid only when a signal appears in the control line.

According to the result from the analysis of multiple sequence alignment, the gene sequence of the large T-antigen contained the highest number of variable positions at which the type of nucleotide in JCV and SV40 simultaneously differs from that in BKV in the consensus sequence at the same sequence position in BKV, JCV, and SV40. Thus, we synthesized a 50-base DNA fragment derived from this region as the target sequence for verification. This fragment was also used to design a 23-base DP and 22-base CP. The number of variable positions in the target DNA sequence of JCV and SV40 was five. Table 1 details the results.

### 3.2. Conjugate Stability

Figure 2 presents the results of the stability analysis of AuNP solutions before and after conjugation. The conjugation of AuNPs and DP at 0.1 M NaCl turned AuNP solutions from cherry red to dark red with no visual aggregates (Figure 2a), although a similar state of affairs was observed in the group of AuNPs coated with only BSA. To characterize the observed results, UV–VIS spectra analyses were performed (Figure 2b). In the spectra results, compared with AuNPs, the AuNP–BSA and AuNP–DP conjugates had a slight red shift in the surface plasmon peak from 525 to 529 nm and 525 to 531 nm, respectively; this was consistent with the color differences.

### 3.3. Assay Optimization

Figure 3 presents the results of optimization assays of AuNP–DP and CP (*n* = 5). To achieve optimal strip performance, we first adjusted the quantity of AuNP–DP (Figure 3a) to determine the optimal level. The evaluation results using 1 OD, 2 OD, and 3 OD AuNP–DP; 100 μM CP; and 50 nM, 100 nM, and 1000 nM target DNA revealed that the relative signal intensity of the AuNP–DP 2 OD group was considerably higher than that of the AuNP–DP 1 OD group (*p* < 0.05 or *p* < 0.01). When the quantity of AuNP–DP was increased to 3 OD, the relative signal intensity decreased significantly relative to that of the 2 OD group (*p* < 0.01). Subsequently, CP concentration was adjusted for optimization (Figure 3b). The results of the evaluation using 50 μM, 100 μM, and 200 μM CP; 2 OD AuNP–DP; and 50 nM, 100 nM, and 1000 nM target DNA indicated that the relative signal intensity of the 100 μM CP group was significantly stronger than those of the 50 μM and 200 μM CP groups (*p* < 0.05 or *p* < 0.01). Figure 4 details the effect on relative signal intensity from the terminus positions of thiol and biotin incorporated at the probe sequence (*n* = 5). AuNP–DP and CP with thiol and biotin incorporated at the 3’ or 5’ terminus were paired to form four combinations (Figure 4a). Analyses were then conducted using 50 nM target DNA, 2 OD AuNP–DP, and 100 μM CP (Figure 4b,c). In the results, the relative signal intensity of the 5’S-3’B probe group was significantly higher than that of other groups (*p* < 0.01).

### 3.4. Sensitivity and Specificity Assays

Sensitivity assays were performed using the optimal quantity of AuNP–DP and CP. Figure 5 presents the test results of the developed strip in the presence of various DNA concentrations (*n* = 3). The detection limit of the developed strip was 5 nM (Figure 5a). The DNA concentration at which a visual signal was detectable was 50 nM, which can be considered the threshold for DNA visual detection. The linear range of detection was 5–1500 nM (Figure 5b). Figure 6 presents the results of the specificity assay of the developed strip (*n* = 3), in which nonspecific binding did not occur, thus indicating that the developed strip was highly specific. At 1500 nM of target DNA, positive results were obtained only in the presence of BKV (*p* < 0.01).

### 3.5. Practicability of the Developed Lateral Flow Strip

Figure 7 demonstrates the linear relationship between relative signal intensity and the amount of plasmid DNA (*n* = 5). To evaluate the practicability of the developed strip in detecting BKV in a natural state, a simulation test was performed using plasmid DNA as the target analyte. The results demonstrated that in the detection of plasmid DNA, the developed strip had a linear detection range of 1 × 10^7^–10^10^ copies/mL.

## 4. Discussion

BKV has garnered heightened attention in recent years due to its high prevalence in the human population. Serological studies have reported that the seroprevalence of BKV peaks at 83%–100% by the age of 10 years [4,5,6]. The route of transmission is not well understood because BKV infections are generally asymptomatic or accompanied by only mild symptoms of upper respiratory infection. Viral DNA has also been detected in the tonsils and respiratory aspirates [26,27], suggesting the possibility of respiratory transmission. The fact that BKV DNA has also been detected in human urine, stool, saliva, and blood samples [28,29,30,31] suggests that BKV may be transmitted through blood transfusion or via the fecal/urine–oral route. Noticeably, scientists have used quantitative PCR to investigate aquatic environments around the world in recent years. They have found that the DNA of BKV can be detected in sewage, surface water, seawater, and even drinking water [32]. They have also speculated that humans are exposed to BKV through their food and water. In 2014, the United States National Institute of Allergy and Infectious Diseases included BKV in its list of Additional Emerging Infectious Diseases/Pathogens [33]. The World Health Organization (WHO) also included BKV in its 2016 Global Water Pathogen Project [34]. BKV infections may be related to cancer risk, as evident by the detection of BKV DNA in cancerous tissue samples of various types, such as colorectal tumors, lymphomas, pancreatic cancer, brain tumors, and prostate cancer [35]. Additionally, studies have reported that BKV further elevates the risk of cancer in solid organ transplant recipients. The incidence of invasive bladder cancer in kidney transplant recipients with and without BKV infection has been noted to be 4.5-fold and 1.7-fold higher than that in the general population, respectively [36,37]. Nevertheless, research results on the correlation between BKV and human malignancy remain inconsistent. Based on ‘‘inadequate evidence’’ in humans and ‘‘sufficient evidence’’ in experimental animals, a WHO International Agency for Cancer Research Monograph Working Group classified BKV in Group 2B as ‘‘possibly carcinogenic to humans’’ [38].

Clinical diseases caused by BKV are extremely rare in immunocompetent individuals. In immunosuppressed individuals, uncontrolled BKV reactivates and replicates, causing serious clinical consequences, particularly in kidney transplant recipients. Approximately 1%–14% of kidney transplant recipients typically develop BKVAN-induced premature renal failure in the first 2 years after their transplant. Without intervention, approximately 90% of kidney transplant recipients with BKVAN exhibit a decline in kidney allograft function, which is followed by an at least 50% graft loss [39]. Even if elevated serum creatinine occurs in early BKVAN [40], the lack of BKV-specific symptoms increases the difficulty of BKVAN diagnosis. Therefore, establishing appropriate detection methods to aid clinical diagnosis and provide early intervention before irreversible damage occurs is imperative for patient disease management. In our study, we used NALFIA to develop a method for BKV detection. To our knowledge, this study is the first to use LFA for BKV detection. AuNPs are an ideal material for biosensor development because they are easy to prepare and feature distinctive optical properties, high biocompatibility, low toxicity, and well-established bioconjugation chemistry. DNA and AuNPs are both negatively charged, necessitating the use of salt (NaCl) in conjugation to circumvent long-range electrostatic repulsion. However, salt also causes AuNP aggregation. Therefore, we first evaluated the colloidal stability of AuNP conjugates after conjugation (Figure 2). In the results, the solution of the BSA-coated AuNP group and the conjugate groups turned a darker color. A slight red shift in the surface plasmon peak was observed in the UV–VIS spectrum. AuNPs have distance-dependent optical properties: Dispersed AuNPs are red, whereas aggregated ones are bluish purple [41].

The resonance wavelength and bandwidth of AuNPs are dependent on a number of factors, such as temperature, the refractive index of the surrounding medium, and particle size and shape [42,43,44]. The slight red shift in our results was attributed to the presence of solvents, BSA, and capping ligands (oligonucleotides), which changed the dielectric constant of the environmental media around the AuNPs. Hence, our results indicated that AuNP conjugates remain well dispersed after conjugation.

To optimize the performance of the developed strip, the quantity of AuNP–DP and CP were adjusted to determine the optimal level (Figure 3). In the results, the relative signal intensity of the 2 OD AuNP–DP group was significantly higher than those of the 1 OD and 3 OD groups. Generally, detection sensitivity increases as the number of strands of DP covering the surface of AuNPs increases. However, studies have indicated that densely packed oligonucleotides on the surface of AuNPs lead to interstrand steric crowding and insufficient free volume, which, in turn, influences hybridization efficiency. Because of crowding, electrostatic repulsive interactions also occur, affecting detection sensitivity [45,46]. In recent years, probes for NALFIA have been designed by adding spacers or conjugating diluent DNA, which is shorter than the sequence of DP, to avoid the aforementioned problems while still maintaining detection sensitivity [47,48]. In our study, optimization adjustments of the quantity of DP were based on AuNPs. However, in our results, the relative signal intensity of the 3 OD AuNP–DP group was significantly lower than that of the 2 OD AuNP–DP group. We surmise that this result was due to the presence of excess AuNP–DP, resulting in steric crowding and electrostatic repulsion, which suppressed hybridization efficiency and in turn influenced relative signal intensity. In evaluating the optimal concentration of CP, the relative signal intensity of the 100 μM CP group was significantly higher than those of other groups. We surmised that the excessively high concentration of CP competes with hybridization complexes to bind to streptavidin. In addition, when analytes migrate through the strip, being driven by capillary force, the CP that binds to streptavidin moves faster because of its smaller molecule and, therefore, it is more likely to occupy the antibody binding sites on the test line before the hybridization complex does. Consequently, relative signal intensity decreased considerably. We next determined the effects on relative signal intensity from the terminus positions of thiol and biotin incorporated at the probe sequence (Figure 4). In our results, the 5’S-3’B probe group had the strongest intensity, probably because the AuNPs and biotin are located on both sides of the hybridization complex, forming a small steric hindrance that increases the likelihood of antibodies on the test and control lines being captured.

We analyzed strip sensitivity and specificity under optimal experimental conditions (Figure 5 and Figure 6). According to the results, the detection limit of the strip was 5 nM. Although 5 nM and 10 nM are not visible to the naked eye, quantitative analysis of the image can still be used to aid data interpretation. The DNA sequences of JCV and SV40 are 75% and 70% homologous to that of BKV, respectively [49,50]. Although the natural host of SV40 is not human, the DNA of this virus can be detected in various human tissue samples [51]. Therefore, SV40 was used as the target analyte for specificity assays. The results revealed that cross-reactivity was not evident; this implies that the developed strip was highly specific. To assess the practicability of the developed strip, a simulation test was performed using plasmid DNA containing the complete genomic sequence of BKV. In the results, the minimum concentration detected was 1 × 10^7^ copies/mL. Existing PCR methods for measuring BKV viral load in urine recommend a load of ≥ 10^7^ copies/mL as the threshold for the development of BKVAN [52]. In future studies, we plan to (1) verify the practicability of the developed strip by using clinical samples and (2) evaluate the performance of the strip on various sample media to determine the feasibility of using the strip in other areas of testing in addition to meeting clinical demands for point-of-care testing.

## 5. Conclusions

In this study, the NALFIA could quickly and easily detect BKV with high specificity in the absence of nucleic acid amplification. Because oligonucleotide probes were used, the cost of analysis was effectively reduced, making our method especially suitable for areas or laboratories with limited resources. The developed immunoassay aids BKV detection and disease management, thereby allowing for early intervention. Our study provides a new option for the clinical monitoring of BKV.

## Figures and Tables

**Figure 1 diagnostics-10-00403-f001:**
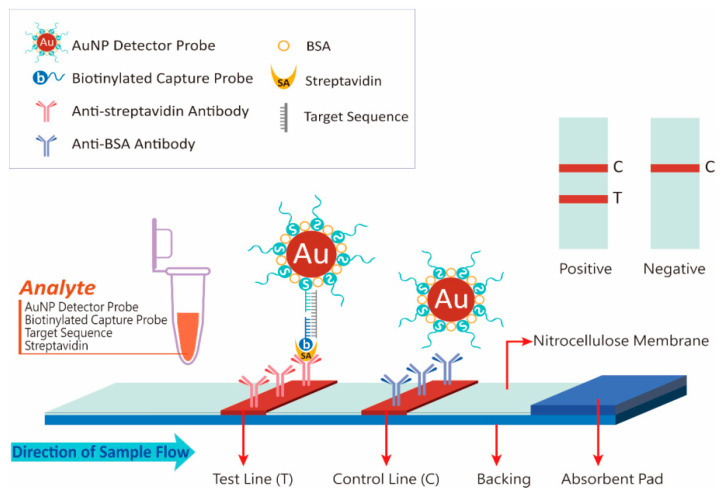
Schematic of the developed lateral flow strip.

**Figure 2 diagnostics-10-00403-f002:**
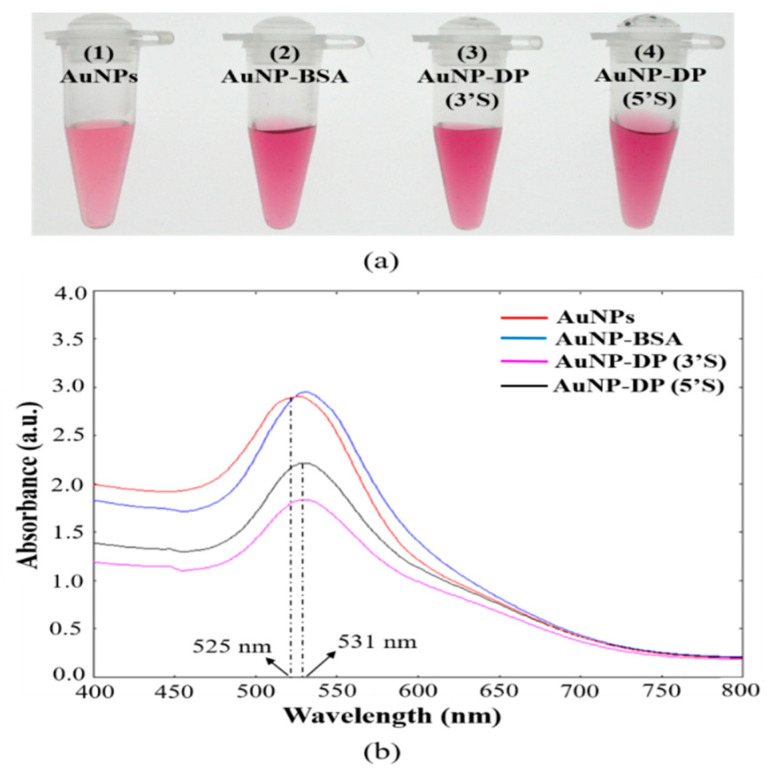
Evaluation of colloidal stability of the gold nanoparticle (AuNP) solutions. (**a**) Photographic images of the AuNP solutions: (1) AuNPs, (2) AuNPs coated with bovine serum albumin (AuNP-BSA), (3) AuNPs conjugated with 3’-thiol modified detector probe [AuNP-DP (3’S)], and AuNPs conjugated with 5’-thiol modified detector probe [AuNP-DP (5’S)]. (**b**) UV–VIS spectra of the solutions from (**a**).

**Figure 3 diagnostics-10-00403-f003:**
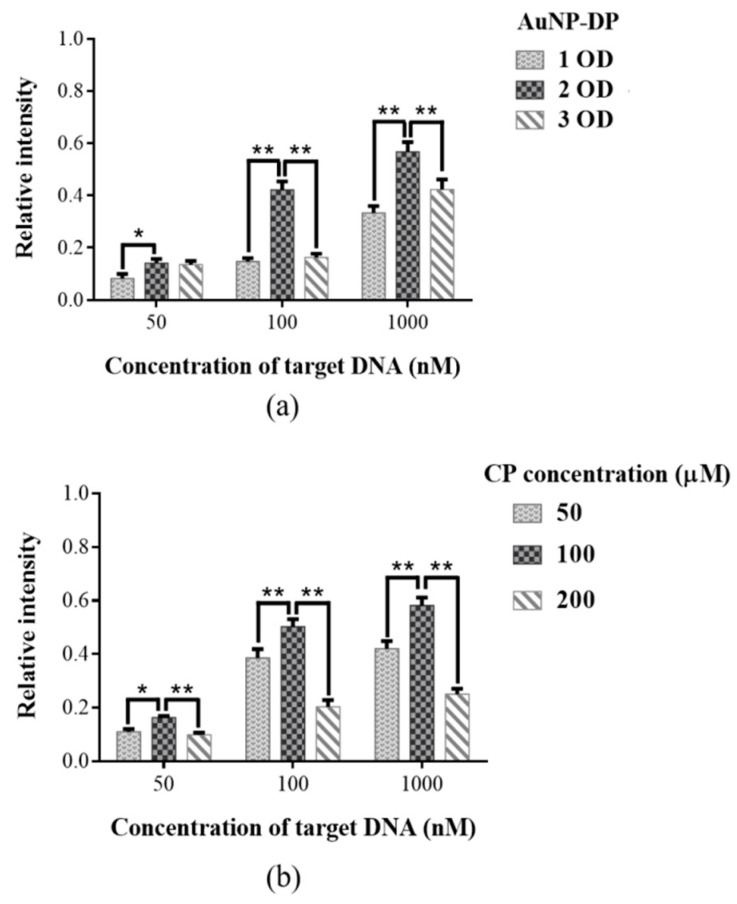
Optimization assays of (**a**) gold nanoparticle–detector probe (AuNP–DP) and (**b**) capture probe (CP); * *p* < 0.05; ** *p* < 0.01; data are presented as the mean and SD.

**Figure 4 diagnostics-10-00403-f004:**
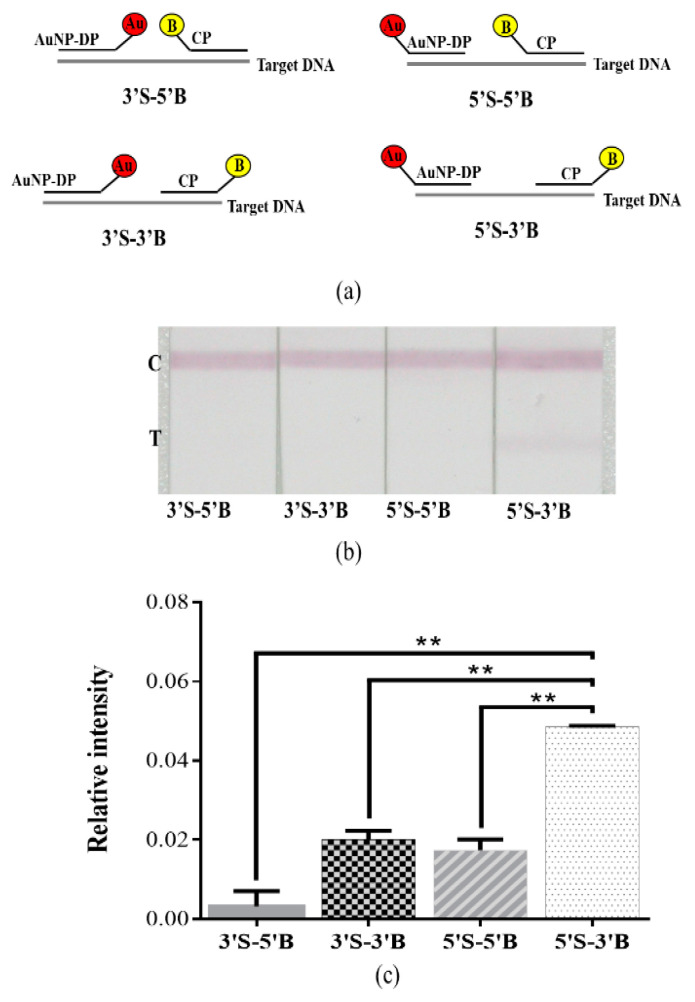
Effect on signal intensity from the terminus positions of thiol and biotin incorporated at the probe sequence. (**a**) Diagram of the relative positions of the termini, which had thiol and biotin attached, on the probe sequence for each combination: 3’S-5’B, 3’S-3’B, 5’S-5’B, and 5’S-3’B. (**b**) Photographic images of the detection of BKV target DNA by the developed lateral flow strip. (**c**) Relative intensity analyses of (**b**). The ** *p* < 0.01; data are presented as the mean and SD.

**Figure 5 diagnostics-10-00403-f005:**
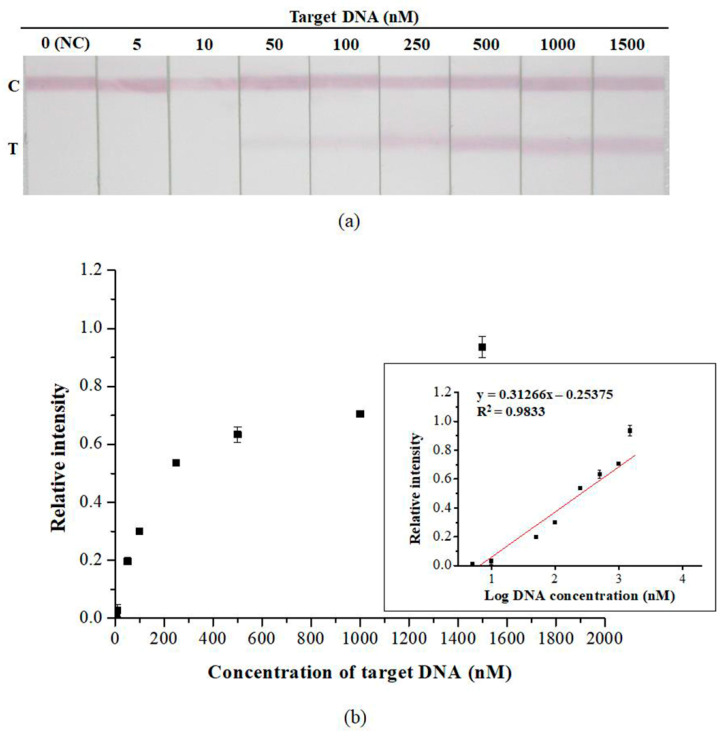
Sensitivity assay of the developed lateral flow strip. (**a**) Photographic images of the developed lateral flow strip in the presence of various DNA concentrations. (**b**) Resulting calibration curve of (**a**). Data are presented as the mean and SD. NC: negative control.

**Figure 6 diagnostics-10-00403-f006:**
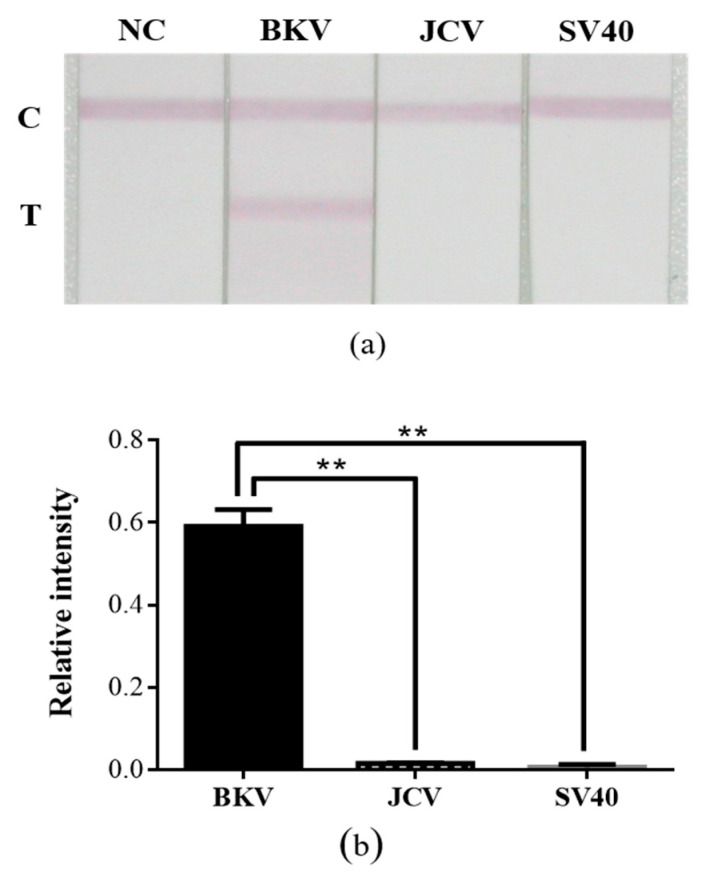
Specificity assay of the developed lateral flow strip. (**a**) Photographic images of the developed lateral flow strip in the presence of the various types of viral target DNA. (**b**) Relative intensity analyses of (**a**). The ** *p* < 0.01; data are presented as the mean and SD. NC: Negative control.

**Figure 7 diagnostics-10-00403-f007:**
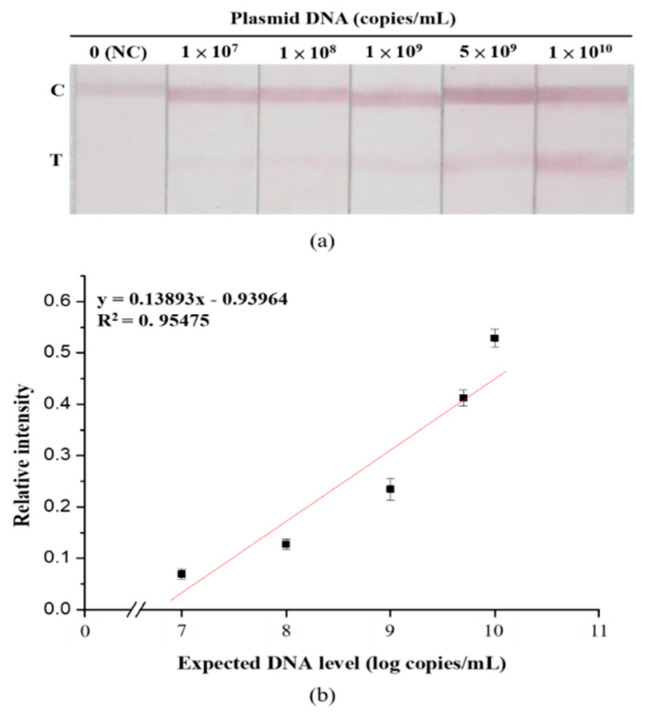
Detectability of the developed lateral flow strip. (**a**) Photographic images of the developed lateral flow strip in the presence of various concentrations of BKV plasmid DNA. (**b**) Resulting calibration curve of (**a**). Data are presented as the mean and SD. NC: Negative control.

**Table 1 diagnostics-10-00403-t001:** Oligonucleotide sequences used for the development of nucleic acid lateral flow immunoassay.

Name	Sequence (5’ to 3’)
Detector probe (5’S)	HS-A_10_-TATGTATGAATAGAGTCTTAGGT
Detector probe (3’S)	TATGTATGAATAGAGTCTTAGGT-A_10_-SH
Capture probe (5’B)	Biotin-A_10_-GAAAGGAAGGTAAGTTGTTAAG
Capture probe (3’B)	GAAAGGAAGGTAAGTTGTTAAG-A_10_-Biotin
BKV target DNA	ATCTTAACA**A**CT**T**ACCTT**C**CTTTCCGACCTAA**GA**CT CTATTCATACATAT
JCV target DNA	CTCCTAACA**C**GT**C**ACCTT**T**CTTTCCGACCTAA**AT**CT TTATTCGTACATAT
SV40 target DNA	GTCTTAACA**C**CT**C**ACCTT**T**CTCTCTAACCTGT**TT**CT CAAATCAAACAGTC

A_10_ is a sequence of 10 adenosine residues and used as a spacer. The bold and underlined bases represent variable nucleotide positions. SH: Thiol group; BKV: BK virus; JCV: JC virus; SV40: Simian virus 40.

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
