# Peer review of "Development of a Nucleic Acid Lateral Flow Immunoassay for the Detection of Human Polyomavirus BK"

_diagnostics, 2020, doi:10.3390/diagnostics10060403_

Round 1

Reviewer 1 Report

Dear Editor,

This paper described a lateral flow assay for the detection of BK virus (BKV). Authors presented results about the performance of this sensor. These data are interesting, I consider that experimental results shown are enough for publishing this paper.

Moreover, the paper is well structured. Therefore, I do recommend considering the publication of this paper after addressing the following minor concerns.

  1. Figure 5: The log calibration plot should show bigger size.
  2. Figure 5: The point 1000 nM in the calibration plot should delete because it is not inside to the lineal adjust.

Sincerely,

                  The reviewer

Author Response

Andreas Kjaer

Editor-in-Chief

Diagnostics (Special issue)- "Trends in Multiplex and Smart Technologies for In Vitro Diagnostics and Point-of-Care Testing"

June 05, 2020

Dear Editor:

Re: Document reference No. diagnostics-820037

Please find attached a revised version of our document “Development of a Nucleic Acid Lateral Flow Immunoassay for the Detection of Human Polyomavirus BK”. We would like to resubmit for publication as an article in Diagnostics (Special issue)- "Trends in Multiplex and Smart Technologies for In Vitro Diagnostics and Point-of-Care Testing".

Your comments and those of the reviewers were highly insightful and enabled us to improve the quality of our document. In the following pages are our responses to each comment from the reviewer(s) as well as your own comments.

Revisions in the text are shown yellow highlights. We hope that our revisions to the document combined with our accompanying responses will be sufficient to render our document suitable for publication in Diagnostics (Special issue)- "Trends in Multiplex and Smart Technologies for In Vitro Diagnostics and Point-of-Care Testing".

We look forward to hearing from you soon.

Yours sincerely,

Cheng-Che Liu, P.h.D

Department of Physiology and Biophysics, Graduate Institute of Physiology, and Institute of Preventive Medicine, National Defense Medical Center, Taipei City, Taiwan (R.O.C.)

Tel.: +886-2-87923100 ext. 18599

Fax: +886-2-87923153

E-Mail: jasonliu@mail.ndmctsgh.edu.tw

Address: No. 161, Sec. 6, Minquan E. Rd., Neihu Dist., Taipei City11490, Taiwan (R.O.C.)

Responses to the comments of Reviewer #1

This paper described a lateral flow assay for the detection of BK virus (BKV). Authors presented results about the performance of this sensor. These data are interesting, I consider that experimental results shown are enough for publishing this paper.

Moreover, the paper is well structured. Therefore, I do recommend considering the publication of this paper after addressing the following minor concerns.

  1. Figure 5: The log calibration plot should show bigger size.

Response: Thank you for your comment. We have revised Figure 5 according to your suggestion (Line 268).

Based on the suggestion of Reviewer #1, we uploaded a revised version of figure 5 for updating.

  1. Figure 5: The point 1000 nM in the calibration plot should delete because it is not inside to the lineal adjust.

Response: We have removed the analysis results regarding the 1000-nM target DNA. The coefficient of determination (R2) has been reduced from 0.9833 to 0.9581. Please refer to the following figure:

Please consider retaining the measurement results regarding the 1000-nM target DNA.

References:

  1. Hakkinen, H. The gold-sulfur interface at the nanoscale. Nat Chem 2012, 4, 443-455, doi:10.1038/nchem.1352.
  2. Zhao, W.; Lin, L.; Hsing, I.M. Rapid synthesis of DNA-functionalized gold nanoparticles in salt solution using mononucleotide-mediated conjugation. Bioconjug Chem 2009, 20, 1218-1222, doi:10.1021/bc900080p.
  3. Hu, J.; Wang, L.; Li, F.; Han, Y.L.; Lin, M.; Lu, T.J.; Xu, F. Oligonucleotide-linked gold nanoparticle aggregates for enhanced sensitivity in lateral flow assays. Lab Chip 2013, 13, 4352-4357, doi:10.1039/c3lc50672j.
  4. Li, F.; Zhang, H.; Dever, B.; Li, X.F.; Le, X.C. Thermal stability of DNA functionalized gold nanoparticles. Bioconjug Chem 2013, 24, 1790-1797, doi:10.1021/bc300687z.
  5. Dharanivasan, G.; Mohammed Riyaz, S.U.; Michael Immanuel Jesse, D.; Raja Muthuramalingam, T.; Rajendran, G.; Kathiravan, K. DNA templated self-assembly of gold nanoparticle clusters in the colorimetric detection of plant viral DNA using a gold nanoparticle conjugated bifunctional oligonucleotide probe. RSC Advances 2016, 6, 11773-11785, doi:10.1039/c5ra25559g.
  6. Choi, J.R.; Liu, Z.; Hu, J.; Tang, R.; Gong, Y.; Feng, S.; Ren, H.; Wen, T.; Yang, H.; Qu, Z., et al. Polydimethylsiloxane-Paper Hybrid Lateral Flow Assay for Highly Sensitive Point-of-Care Nucleic Acid Testing. Anal Chem 2016, 88, 6254-6264, doi:10.1021/acs.analchem.6b00195.
  7. Herdt, A.R.; Drawz, S.M.; Kang, Y.; Taton, T.A. DNA dissociation and degradation at gold nanoparticle surfaces. Colloids Surf B Biointerfaces 2006, 51, 130-139, doi:10.1016/j.colsurfb.2006.06.006.
  8. Jain, P.K.; Lee, K.S.; El-Sayed, I.H.; El-Sayed, M.A. Calculated absorption and scattering properties of gold nanoparticles of different size, shape, and composition: applications in biological imaging and biomedicine. J Phys Chem B 2006, 110, 7238-7248, doi:10.1021/jp057170o.
  9. Muhlig, S.; Rockstuhl, C.; Yannopapas, V.; Burgi, T.; Shalkevich, N.; Lederer, F. Optical properties of a fabricated self-assembled bottom-up bulk metamaterial. Opt Express 2011, 19, 9607-9616, doi:10.1364/OE.19.009607.
  10. O’Farrell, B. Lateral Flow Immunoassay Systems: Evolution from the Current State of the Art to the Next Generation of Highly Sensitive, Quantitative Rapid Assays. In The Immunoassay Handbook, 4th ed.; Wild, D., Ed. Elsevier Science: 2013; 10.1016/b978-0-08-097037-0.00007-5pp. 89-107.
  11. Rong-Hwa, S.; Shiao-Shek, T.; Der-Jiang, C.; Yao-Wen, H. Gold nanoparticle-based lateral flow assay for detection of staphylococcal enterotoxin B. Food Chemistry 2010, 118, 462-466, doi:10.1016/j.foodchem.2009.04.106.
  12. Rodriguez, M.O.; Covian, L.B.; Garcia, A.C.; Blanco-Lopez, M.C. Silver and gold enhancement methods for lateral flow immunoassays. Talanta 2016, 148, 272-278, doi:10.1016/j.talanta.2015.10.068.
  13. Liu, C.C.; Yu, J.S.; Wang, P.J.; Hsiao, Y.C.; Liu, C.H.; Chen, Y.C.; Lai, P.F.; Hsu, C.P.; Fann, W.C.; Lin, C.C. Development of sandwich ELISA and lateral flow strip assays for diagnosing clinically significant snakebite in Taiwan. PLoS Negl Trop Dis 2018, 12, e0007014, doi:10.1371/journal.pntd.0007014.
  14. Taton, T.A. Preparation of gold nanoparticle-DNA conjugates. Curr Protoc Nucleic Acid Chem 2002, Chapter 12, Unit 12 12, doi:10.1002/0471142700.nc1202s09.
  15. Hurst, S.J.; Lytton-Jean, A.K.; Mirkin, C.A. Maximizing DNA loading on a range of gold nanoparticle sizes. Anal Chem 2006, 78, 8313-8318, doi:10.1021/ac0613582.
  16. Ramos, E.; Drachenberg, C.B.; Wali, R.; Hirsch, H.H. The decade of polyomavirus BK-associated nephropathy: state of affairs. Transplantation 2009, 87, 621-630, doi:10.1097/TP.0b013e318197c17d.
  17. Pang, X.L.; Doucette, K.; LeBlanc, B.; Cockfield, S.M.; Preiksaitis, J.K. Monitoring of polyomavirus BK virus viruria and viremia in renal allograft recipients by use of a quantitative real-time PCR assay: one-year prospective study. J Clin Microbiol 2007, 45, 3568-3573, doi:10.1128/JCM.00655-07.
  18. Hirsch, H.H.; Knowles, W.; Dickenmann, M.; Passweg, J.; Klimkait, T.; Mihatsch, M.J.; Steiger, J. Prospective study of polyomavirus type BK replication and nephropathy in renal-transplant recipients. N Engl J Med 2002, 347, 488-496, doi:10.1056/NEJMoa020439.
  19. Bohl, D.L.; Brennan, D.C. BK virus nephropathy and kidney transplantation. Clin J Am Soc Nephrol 2007, 2 Suppl 1, S36-46, doi:10.2215/CJN.00920207.
  20. Heritage, J.; Chesters, P.M.; McCance, D.J. The persistence of papovavirus BK DNA sequences in normal human renal tissue. J Med Virol 1981, 8, 143-150, doi:10.1002/jmv.1890080208.
  21. Lalani, T.; Tisdale, M.D.; Liu, J.; Mitra, I.; Philip, C.; Odundo, E.; Reyes, F.; Simons, M.P.; Fraser, J.A.; Hutley, E., et al. Comparison of stool collection and storage on Whatman FTA Elute cards versus frozen stool for enteropathogen detection using the TaqMan Array Card PCR assay. PLoS One 2018, 13, e0202178, doi:10.1371/journal.pone.0202178.
  22. Hashimoto, M.; Bando, M.; Kido, J.I.; Yokota, K.; Mita, T.; Kajimoto, K.; Kataoka, M. Nucleic acid purification from dried blood spot on FTA Elute Card provides template for polymerase chain reaction for highly sensitive Plasmodium detection. Parasitol Int 2019, 73, 101941, doi:10.1016/j.parint.2019.101941.

Reviewer 2 Report

The manuscript by Huang et al. entitled Development of a Nucleic Acid Lateral Flow Immunoassay for the Detection of Human Polyomavirus BK is describing a development of method for detection of BK virus that may cause serious complications especially in the case of immunosuppressed individuals. Developed method is based on nucleic acid lateral flow immunoassay and offers great potential for BKV detection in point-of-care settings, thus competing with the methods used so far (as PCR).

In introduction of the article the problematics of BKV and necessity of development of fast and simply-to-use diagnostic assay is explained in detail. Also, benefits and disadvantages of suggested diagnostic method are discussed. Even though the study is carefully designed and performed, in my opinion, there are several major issues that need to be addressed.

  • From results in Figure 2 is seen that after interaction of AuNPs with DP change of AuNP absorbance occurs, this according authors confirms that conjugate was crated. However, in discussion authors state that "The resonance wavelength and bandwidth of AuNPs are dependent on a number of factors, such as temperature, the refractive index of the surrounding medium, and particle size and shape." The confirmation of the creation of conjugates by some other analytical method (such HPLC, CE, etc.) would be suitable.
  • The conjugate stability from data in Figure 2 is not clear. Has it been determined for how long the conjugate is stable and functional?
  • The antibody immobilization process is not properly described and neith stability of the antibody nor it quantity was determined. This is sufficient for YES/NO test but not for quantitative assays
  • In the discussion, authors state that "Generally, detection sensitivity increases as the number of strands of DP covering the surface of AuNPs increases. However, studies have indicated that densely packed oligonucleotides on the surface of AuNPs lead to interstream steric crowding and insufficient free volume, which, in turn, influences hybridization efficiency." It implies that the ratio of AuNPs and DP has a huge influence on the selectivity of AuNPs-DP conjugate. Was the ratio of AuNPs and DP optimized?
  • In discussion authors mentioned that they plan to use developed strip to determine the BK virus in clinical samples. In what amount the BK virus is approximately present in the clinical samples? And will be detection ability of the developed strips sufficient?
  • The authors state that the price of the developed strip is lower than the price of currently used methods. How much does the detection by the developed strip cost?
  • Author should demonstrate real sample analysis. Does it involve isolation of nucleic acid or amplification? This would limit the applicability

Author Response

Andreas Kjaer

Editor-in-Chief

Diagnostics (Special issue)- "Trends in Multiplex and Smart Technologies for In Vitro Diagnostics and Point-of-Care Testing"

June 05, 2020

Dear Editor:

Re: Document reference No. diagnostics-820037

Please find attached a revised version of our document “Development of a Nucleic Acid Lateral Flow Immunoassay for the Detection of Human Polyomavirus BK”. We would like to resubmit for publication as an article in Diagnostics (Special issue)- "Trends in Multiplex and Smart Technologies for In Vitro Diagnostics and Point-of-Care Testing".

Your comments and those of the reviewers were highly insightful and enabled us to improve the quality of our document. In the following pages are our responses to each comment from the reviewer(s) as well as your own comments.

Revisions in the text are shown yellow highlights. We hope that our revisions to the document combined with our accompanying responses will be sufficient to render our document suitable for publication in Diagnostics (Special issue)- "Trends in Multiplex and Smart Technologies for In Vitro Diagnostics and Point-of-Care Testing".

We look forward to hearing from you soon.

Yours sincerely,

Cheng-Che Liu, P.h.D

Department of Physiology and Biophysics, Graduate Institute of Physiology, and Institute of Preventive Medicine, National Defense Medical Center, Taipei City, Taiwan (R.O.C.)

Tel.: +886-2-87923100 ext. 18599

Fax: +886-2-87923153

E-Mail: jasonliu@mail.ndmctsgh.edu.tw

Address: No. 161, Sec. 6, Minquan E. Rd., Neihu Dist., Taipei City11490, Taiwan (R.O.C.)

Responses to the comments of Reviewer #2

The manuscript by Huang et al. entitled Development of a Nucleic Acid Lateral Flow Immunoassay for the Detection of Human Polyomavirus BK is describing a development of method for detection of BK virus that may cause serious complications especially in the case of immunosuppressed individuals. Developed method is based on nucleic acid lateral flow immunoassay and offers great potential for BKV detection in point-of-care settings, thus competing with the methods used so far (as PCR).

In introduction of the article the problematics of BKV and necessity of development of fast and simply-to-use diagnostic assay is explained in detail. Also, benefits and disadvantages of suggested diagnostic method are discussed. Even though the study is carefully designed and performed, in my opinion, there are several major issues that need to be addressed.

  1. From results in Figure 2 is seen that after interaction of AuNPs with DP change of AuNP absorbance occurs, this according authors confirms that conjugate was crated. However, in discussion authors state that "The resonance wavelength and bandwidth of AuNPs are dependent on a number of factors, such as temperature, the refractive index of the surrounding medium, and particle size and shape." The confirmation of the creation of conjugates by some other analytical method (such HPLC, CE, etc.) would be suitable.

The conjugate stability from data in Figure 2 is not clear. Has it been determined for how long the conjugate is stable and functional?

Response: The affinity between AuNPs and the thiol group is strong; hence, molecules with thiol groups are quickly absorbed by AuNPs to form self-assembled monolayers [1]. Accordingly, the most popular and robust approach of attaching DNA to AuNPs is by using thiol-modified DNA. Various methods can be used to analyze the conjugate stability. UV–vis spectrophotometry is one such common method [2-6]. AuNP conjugates that were functionalized by oligonucleotides are highly stable under high salinity (in our study, the conjugate remained stable in a 0.3 M NaCl solution) and are tolerant to temperature <70°C. A study verified that in environments of >70°C, thiol-terminated DNA may become detached from the surface of AuNPs [7]. High temperatures also change the shape and size of metal nanoparticles because of Ostwald ripening [7]. AuNPs of different shapes and sizes generate considerably different surface plasmon peaks. A large nanoparticle size leads to a long surface plasmon peak wavelength. For example, a 20-nm gold nanospheres exhibits a surface plasmon peak of 521 nm. When the particle size increases to 80 nm, the peak appears at 549 nm [8]. When the distance between particles is reduced because of particle aggregation, the surface plasmon peak becomes broader and shifts toward a long wavelength [9]. According to these results, extinction spectra can be used to determine the stability of AuNP–DP conjugates.

In an ongoing study that is expected to span 1 year, we store the developed strip, AuNP–DP, and CP at 4°C. Repetitions of three trials are performed every few days on the 1000-nM target DNA. This study has been conducted for 50 days. The relative standard deviation of the newly prepared sample and that stored for 50 days are 7.88% and 5.09%, respectively. This indicates that the developed strip and AuNP–DP conjugates are highly stable. The complete stability analysis results will be presented in the next article to be submitted in Diagnostics after the clinical samples are analyzed.

  1. The antibody immobilization process is not properly described and neith stability of the antibody nor it quantity was determined. This is sufficient for YES/NO test but not for quantitative assays

Response: Binding between the antibody and nitrocellulose membrane was induced through electrostatic interaction. The amount of liquid absorbed by every unit area of the membrane was affected by material-related factors; hence, this amount was not quantifiable. These factors included hydration of the membrane, the pore size of the membrane, membrane surface characteristics (smoothness and dust), fluid factors including viscosity and protein concentration, and environmental factors such as the ambient relative humidity of the striping process area. Membranes used in immunochromatographic applications have surface area ratios in the range of 50 to 200. For IgG, the loading capacity is approximately 1 mg/cm2. If the surface area ratio of the membrane is 50-200, then the approximate IgG binding capacity is 50–200 mg/cm2. A typical test line is 1 mm wide. If the strip is 1 cm wide, the amount of bound antibody is 5–20 mg (0.1 cm width × 1 cm length = 0.1 cm2). This amount is 10–100 fold greater than required for most assays. Therefore, protein-binding capacity is typically not an issue in test design [10]. The amount of fluid dispensed from the dispenser is quantitative, as defined by the accuracy of the pump used. Therefore, the concentration of antibodies used to prepare and the dispensing rate are two parameters that mentioned when describing the preparation of the control and test lines [11-13].

Regarding antibody stability, please refer to the aforementioned description of stability analysis.

  1. 3. In the discussion, authors state that "Generally, detection sensitivity increases as the number of strands of DP covering the surface of AuNPs increases. However, studies have indicated that densely packed oligonucleotides on the surface of AuNPs lead to interstream steric crowding and insufficient free volume, which, in turn, influences hybridization efficiency." It implies that the ratio of AuNPs and DP has a huge influence on the selectivity of AuNPs-DP conjugate. Was the ratio of AuNPs and DP optimized?

Response: Thank you for your comment. The density of the DP covering the AuNP surface was an essential factor that affected detection sensitivity. Therefore, the method used to conjugate the DP with AuNPs in this study was based on two major articles, one was published by Taton et al. (2002) in Current Protocols in Nucleic Acid Chemistry, and the other was published by Hurst et al. (2006) in Analytical Chemistry [14,15]. Specifically, 10 mL of 100-mM thiol-modified DP was loaded into 500 mL of 35-nm colloidal gold until the approximate DP density on the nanoparticle surface reached 0.576 nmol/cm2. Hybridizations were performed to verify the conjugation of the AuNP–DP. The LOD experiment in our study revealed that the minimum detection limit of the AuNP–DP conjugate was between 10 and 50 nM, which verified the feasibility of the maximum density estimated according to the references.

  1. 4. In discussion authors mentioned that they plan to use developed strip to determine the BK virus in clinical samples. In what amount the BK virus is approximately present in the clinical samples? And will be detection ability of the developed strips sufficient?

Response: After the BKV was reactivated in the patient, viruria occurred for several weeks, followed by viremia [16]. According to the clinical sample analysis of multiple organ transplantation centers globally, quantitative PCR is recommended for analyzing blood and urine samples. The thresholds of BKVAN progression are a DNA load of ≥ 1 × 107 copies/mL in the urine sample and ≥ 104 copies/mL in the plasma sample [17-19]. BKV DNA in the host is mainly free form rather than integrated into the host DNA [20]. Therefore, we used pBKV plasmid DNA to perform simulation testing. The minimum detection concentration was 1 × 107 copies/mL. In the very beginning, we would like to apply our developed strip to the urine sample. The LFA technique mainly answers qualitative questions. If the results of the developed strip analysis are positive, the patient requires a further detailed examination. 

  1. 5. The authors state that the price of the developed strip is lower than the price of currently used methods. How much does the detection by the developed strip cost?

Response: According to our estimation, before this strip production method becomes commercialized, the cost of a developed strip is approximately USD2.81 per unit. The price of a BKV DNA test kit with CE-IVD certification, RealStar BKV PCR kit 1.0 (altona Diagnostics GmbH, Hamburg, Germany), that utilizes real-time quantitative PCR detection is USD30.78 per reaction in Taiwan.

  1. Author should demonstrate real sample analysis. Does it involve isolation of nucleic acid or amplification? This would limit the applicability

Response: Our next step is verification by analyzing clinical samples. We received IRB approval (as shown below) soon after we submitted this article. Therefore, we are currently collecting clinical samples, and the analysis results will be presented in our next article to be published. Our proposed method involves nucleic acid separation but does not require nucleic acid amplification. In practice, nucleic acid separation in clinical samples can be achieved using commercial products such as the Whatman FTA Card (GE Healthcare Life Sciences, Marlborough, MA, USA) to separate nucleic acids within 30 min. This satisfies the needs for onsite analysis. In recent years, published studies have used this approach to separate nucleic acid and perform analysis [6,21,22].

,

References:

  1. Hakkinen, H. The gold-sulfur interface at the nanoscale. Nat Chem 2012, 4, 443-455, doi:10.1038/nchem.1352.
  2. Zhao, W.; Lin, L.; Hsing, I.M. Rapid synthesis of DNA-functionalized gold nanoparticles in salt solution using mononucleotide-mediated conjugation. Bioconjug Chem 2009, 20, 1218-1222, doi:10.1021/bc900080p.
  3. Hu, J.; Wang, L.; Li, F.; Han, Y.L.; Lin, M.; Lu, T.J.; Xu, F. Oligonucleotide-linked gold nanoparticle aggregates for enhanced sensitivity in lateral flow assays. Lab Chip 2013, 13, 4352-4357, doi:10.1039/c3lc50672j.
  4. Li, F.; Zhang, H.; Dever, B.; Li, X.F.; Le, X.C. Thermal stability of DNA functionalized gold nanoparticles. Bioconjug Chem 2013, 24, 1790-1797, doi:10.1021/bc300687z.
  5. Dharanivasan, G.; Mohammed Riyaz, S.U.; Michael Immanuel Jesse, D.; Raja Muthuramalingam, T.; Rajendran, G.; Kathiravan, K. DNA templated self-assembly of gold nanoparticle clusters in the colorimetric detection of plant viral DNA using a gold nanoparticle conjugated bifunctional oligonucleotide probe. RSC Advances 2016, 6, 11773-11785, doi:10.1039/c5ra25559g.
  6. Choi, J.R.; Liu, Z.; Hu, J.; Tang, R.; Gong, Y.; Feng, S.; Ren, H.; Wen, T.; Yang, H.; Qu, Z., et al. Polydimethylsiloxane-Paper Hybrid Lateral Flow Assay for Highly Sensitive Point-of-Care Nucleic Acid Testing. Anal Chem 2016, 88, 6254-6264, doi:10.1021/acs.analchem.6b00195.
  7. Herdt, A.R.; Drawz, S.M.; Kang, Y.; Taton, T.A. DNA dissociation and degradation at gold nanoparticle surfaces. Colloids Surf B Biointerfaces 2006, 51, 130-139, doi:10.1016/j.colsurfb.2006.06.006.
  8. Jain, P.K.; Lee, K.S.; El-Sayed, I.H.; El-Sayed, M.A. Calculated absorption and scattering properties of gold nanoparticles of different size, shape, and composition: applications in biological imaging and biomedicine. J Phys Chem B 2006, 110, 7238-7248, doi:10.1021/jp057170o.
  9. Muhlig, S.; Rockstuhl, C.; Yannopapas, V.; Burgi, T.; Shalkevich, N.; Lederer, F. Optical properties of a fabricated self-assembled bottom-up bulk metamaterial. Opt Express 2011, 19, 9607-9616, doi:10.1364/OE.19.009607.
  10. O’Farrell, B. Lateral Flow Immunoassay Systems: Evolution from the Current State of the Art to the Next Generation of Highly Sensitive, Quantitative Rapid Assays. In The Immunoassay Handbook, 4th ed.; Wild, D., Ed. Elsevier Science: 2013; 10.1016/b978-0-08-097037-0.00007-5pp. 89-107.
  11. Rong-Hwa, S.; Shiao-Shek, T.; Der-Jiang, C.; Yao-Wen, H. Gold nanoparticle-based lateral flow assay for detection of staphylococcal enterotoxin B. Food Chemistry 2010, 118, 462-466, doi:10.1016/j.foodchem.2009.04.106.
  12. Rodriguez, M.O.; Covian, L.B.; Garcia, A.C.; Blanco-Lopez, M.C. Silver and gold enhancement methods for lateral flow immunoassays. Talanta 2016, 148, 272-278, doi:10.1016/j.talanta.2015.10.068.
  13. Liu, C.C.; Yu, J.S.; Wang, P.J.; Hsiao, Y.C.; Liu, C.H.; Chen, Y.C.; Lai, P.F.; Hsu, C.P.; Fann, W.C.; Lin, C.C. Development of sandwich ELISA and lateral flow strip assays for diagnosing clinically significant snakebite in Taiwan. PLoS Negl Trop Dis 2018, 12, e0007014, doi:10.1371/journal.pntd.0007014.
  14. Taton, T.A. Preparation of gold nanoparticle-DNA conjugates. Curr Protoc Nucleic Acid Chem 2002, Chapter 12, Unit 12 12, doi:10.1002/0471142700.nc1202s09.
  15. Hurst, S.J.; Lytton-Jean, A.K.; Mirkin, C.A. Maximizing DNA loading on a range of gold nanoparticle sizes. Anal Chem 2006, 78, 8313-8318, doi:10.1021/ac0613582.
  16. Ramos, E.; Drachenberg, C.B.; Wali, R.; Hirsch, H.H. The decade of polyomavirus BK-associated nephropathy: state of affairs. Transplantation 2009, 87, 621-630, doi:10.1097/TP.0b013e318197c17d.
  17. Pang, X.L.; Doucette, K.; LeBlanc, B.; Cockfield, S.M.; Preiksaitis, J.K. Monitoring of polyomavirus BK virus viruria and viremia in renal allograft recipients by use of a quantitative real-time PCR assay: one-year prospective study. J Clin Microbiol 2007, 45, 3568-3573, doi:10.1128/JCM.00655-07.
  18. Hirsch, H.H.; Knowles, W.; Dickenmann, M.; Passweg, J.; Klimkait, T.; Mihatsch, M.J.; Steiger, J. Prospective study of polyomavirus type BK replication and nephropathy in renal-transplant recipients. N Engl J Med 2002, 347, 488-496, doi:10.1056/NEJMoa020439.
  19. Bohl, D.L.; Brennan, D.C. BK virus nephropathy and kidney transplantation. Clin J Am Soc Nephrol 2007, 2 Suppl 1, S36-46, doi:10.2215/CJN.00920207.
  20. Heritage, J.; Chesters, P.M.; McCance, D.J. The persistence of papovavirus BK DNA sequences in normal human renal tissue. J Med Virol 1981, 8, 143-150, doi:10.1002/jmv.1890080208.
  21. Lalani, T.; Tisdale, M.D.; Liu, J.; Mitra, I.; Philip, C.; Odundo, E.; Reyes, F.; Simons, M.P.; Fraser, J.A.; Hutley, E., et al. Comparison of stool collection and storage on Whatman FTA Elute cards versus frozen stool for enteropathogen detection using the TaqMan Array Card PCR assay. PLoS One 2018, 13, e0202178, doi:10.1371/journal.pone.0202178.
  22. Hashimoto, M.; Bando, M.; Kido, J.I.; Yokota, K.; Mita, T.; Kajimoto, K.; Kataoka, M. Nucleic acid purification from dried blood spot on FTA Elute Card provides template for polymerase chain reaction for highly sensitive Plasmodium detection. Parasitol Int 2019, 73, 101941, doi:10.1016/j.parint.2019.101941.

Round 2

Reviewer 2 Report

The manuscript is suitable for publication